# Spatial ecology of *Haemophilus and Aggregatibacter* in the human oral cavity

Jonathan J. Giacomini,[1] Julian Torres-Morales,[1] Jonathan Tang,[1] Floyd E. Dewhirst,[1,2] Gary G. Borisy,[1] Jessica L. Mark Welch[1,3]

ABSTRACT   *Haemophilus* and *Aggregatibacter* are two of the most common bacterial genera in the human oral cavity, encompassing both commensals and pathogens of substantial ecological and medical significance. In this study, we conducted a metapangenomic analysis of oral *Haemophilus* and *Aggregatibacter* species to uncover genomic diversity, phylogenetic relationships, and habitat specialization within the human oral cavity. Using three metrics—pangenomic gene content, phylogenomics, and average nucleotide identity (ANI)—we first identified distinct species and sub-species groups among these genera. Mapping of metagenomic reads then revealed clear patterns of habitat specialization, such as *Aggregatibacter* species predominantly in dental plaque, a distinctive *Haemophilus parainfluenzae* sub-species group on the tongue dorsum, and *H*. sp. HMT-036 predominantly in keratinized gingiva and buccal mucosa. In addition, we found that supragingival plaque samples contained predominantly only one out of the three taxa, *H. parainfluenzae*, *Aggregatibacter aphrophilus*, and *A*. sp. HMT-458, suggesting independent niches or a competitive relationship. Functional analyses revealed the presence of key metabolic genes, such as oxaloacetate decarboxylase, correlated with habitat specialization, suggesting metabolic versatility as a driving force. Additionally, heme synthesis distinguishes *H*. sp. HMT-036 from closely related *Haemophilus haemolyticus*, suggesting that the availability of micronutrients, particularly iron, was important in the evolutionary ecology of these species. Overall, our study exemplifies the power of metapangenomics to identify factors that may affect ecological interactions within microbial communities, including genomic diversity, habitat specialization, and metabolic versatility.

IMPORTANCE   Understanding the microbial ecology of the mouth is essential for comprehending human physiology. This study employs metapangenomics to reveal that various *Haemophilus* and *Aggregatibacter* species exhibit distinct ecological preferences within the oral cavity of healthy individuals, thereby supporting the site-specialist hypothesis. Additionally, it was observed that the gene pool of different *Haemophilus* species correlates with their ecological niches. These findings shed light on the significance of key metabolic functions in shaping microbial distribution patterns and interspecies interactions in the oral ecosystem.

KEYWORDS   *Haemophilus*, *Aggregatibacter*, pangenomics, metagenomics, metapangenomics, tropism, site-specialists

The human oral microbiome comprises bacteria that have specifically evolved to inhabit the oral environment. Bacteria that dominate in the human mouth are generally rare in the gut or on the skin, and vice versa (1–3), and many oral species have strong preferences for specific habitats within the mouth, such as the hard surfaces of teeth or soft tissues of the tongue (4–6). These habitats, together with disturbances from

Editor Justin R. Kaspar, The Ohio State University College of Dentistry, Columbus, Ohio, USA

Ad Hoc Peer Reviewer Clifford J. Beall, The Ohio State University College of Dentistry, Columbus, Ohio, USA

Address correspondence to Jonathan J. Giacomini, jonjgiacomini@gmail.com.

The authors declare no conflict of interest.

See the funding table on p. 14.

oral hygiene and dietary practices, shape the distribution, interactions, and specializations of bacterial taxa, resulting in distinct microbial profiles tailored to each niche (7).

Among the numerous bacterial genera found in the human oral cavity, *Haemophilus* and *Aggregatibacter* are two of the most common, accounting for a significant proportion of bacteria at most oral sites (7–9). These genera, belonging to the family Pasteurellaceae, encompass both commensals and pathogens with substantial ecological and medical significance (10). For example, *Haemophilus parainfluenzae* is one of the most abundant and transcriptionally active species in the human oral cavity (11). Conversely, *Haemophilus influenzae* is a significant opportunistic pathogen associated with various respiratory infections and is renowned for its role in numerous scientific advancements (10). Certain species of *Aggregatibacter*, such as *Actinobacillus actinomycetemcomitans*, have been strongly linked to periodontitis, a chronic inflammatory disease involving the loss of connective tissue and bone around teeth (12). Understanding the genomic diversity, habitat specialization, and metabolic versatility of *Haemophilus* and *Aggregatibacter* within natural communities will be important for deconstructing the ecological interactions of the human oral microbiome.

The family Pasteurellaceae has undergone extensive taxonomic expansion and reorganization with the advent of molecular methods. Previously, Pasteurellaceae species were classified into three genera based on limited phenotypic markers: *Haemophilus* for species relying on growth factors in blood and *Pasteurella* or *Actinobacillus* for species without such dependence (13, 14). Certain species, such as *Haemophilus influenzae* and *Haemophilus haemolyticus*, require both haemin (X-factor) and nicotinamide adenine dinucleotide (V-factor), while others, such as *H. parainfluenzae,* only require V-factor. These growth requirements are often used for species differentiation, but recent studies suggest that molecular methods provide more accurate identification (10, 15). Contemporary molecular methods have revealed significant genetic diversity within *Haemophilus,* leading to the establishment of the genus *Aggregatibacter* to accommodate *Actinobacillus actinomycetemcomitans*, *Haemophilus aphrophilus*, and *Haemophilus segnis* as a group primarily associated with humans (16).

Presently, the Human Oral Microbiome Database (HOMD; [www.ehomd.org](www.ehomd.org)) recognizes twelve *Haemophilus* species and four *Aggregatibacter* species. Among them are two recently isolated and unnamed species currently referred to as Human Microbial Taxon (HMT) 036, and HMT-458. Oligotyping using 16S rRNA gene sequences indicated that HMT-458 is a prevalent species present in multiple oral habitats (8). Conversely, oligotyping suggested that HMT-036 is a habitat specialist predominantly found on keratinized gingiva (8). However, drawing conclusions about ecological relationships based solely on short sequences of 16S rRNA is intrinsically limited. The increasing availability of whole genome sequences enables a more reliable analysis of taxonomic relationships and ecological distribution patterns.

In this study, we employ metapangenomics (5, 9), an innovative approach that combines metagenomics and pangenomics to comprehensively analyze the distribution, genetic diversity, and functional roles of *Haemophilus* and *Aggregatibacter* within the human oral microbiome. Specifically, we aim to investigate whether *Haemophilus* and *Aggregatibacter* species exhibit habitat specificity or have a broader range in the human oral cavity. We then aimed to identify core gene functions that may contribute to the oral site preferences of *Haemophilus* and *Aggregatibacter* species. By leveraging genomic and metagenomic information, we identified specific oral niches that provide favorable conditions for the growth of *Haemophilus* and *Aggregatibacter* species, shedding light on their metabolic pathways, their interactions with other members of the oral microbiome, and their overall contribution to the ecology of this microbial ecosystem.

## MATERIALS AND METHODS

The following analyses were conducted primarily using the Anvi'o v7.1 (17) platform with Python v3.7.9 (18). Figures were generated using R v4.1.2 (19) and manually edited using Inkscape v1.2.2 (20).

### *Haemophilus* and *Aggregatibacter* reference genomes

To establish a reference genome collection representing *Haemophilus* and *Aggregatibacter* populations in the human oral microbiome, we obtained publicly available RefSeq genomes from the National Center for Biotechnology Information (NCBI) database (downloaded on August 15, 2022) for *Haemophilus* and *Aggregatibacter* species that were isolated from human hosts. Out of the 1226 genome assemblies obtained, 1018 were identified by the NCBI as *Haemophilus* and 208 as *Aggregatibacter*. Further details about the genomes, such as their genus, species, strain, BioSample, BioProject, isolation host, isolation site, RefSeq status, type strain, disease association, and submitter ID, can be found in Table S1.

We then performed quality control and dereplication steps to ensure that each genome in the collection had a completeness of at least 90%, had a contamination level below 5% as estimated by CheckM (21), and had no more than 98% average nucleotide identity (ANI) with any other genome (22). This process resulted in a set of 188 high-quality reference genomes belonging to 17 species representing the diversity of *Haemophilus* and *Aggregatibacter* found in the human oral microbiome, as well as 14 genomes that were only categorized to the genus level. In total, we used 202 genomes to construct the *Haemophilus* and *Aggregatibacter* pangenome.

### Constructing an oral *Haemophilus* and *Aggregatibacter* pangenome

To construct the pangenome, we adapted previously developed methods (4–6, 9). First, we removed all contigs from each reference genome that were less than 300 nt and replaced non-canonical nucleotide letters with "N. We then converted each genome into an Anvi'o-compatible contigs database using *anvi-gen-contigs-db*. Open reading framesereafter referred to as genes, were identified in each genome using Prodigal (v2.6.3) (23). Functional annotation of genes was achieved using multiple Anvi'o scripts, including *anvi-run-hmms* to find bacterial single-copy genes (Bacteria71 SCG set) (24, 25) with hidden Markov Model (HMM) profiles, *anvi-run-ncbi-cogs* using blastp (v2.10.1+) to annotate with the cluster of orthologous genes (COGs) database (version COG20) (26), and *anvi-run-pfams* and *anvi-run-kegg-kofams* with hmmscan from HMMER (v3.3.1) (27) to functionally annotate with Pfams (v34.0) (28) and KOfams/ KEGG Modules (v97.0) (29), respectively. We then used *anvi-pan-genome* to construct the annotated pangenome using BLASTP to calculate the amino acid-level identity between all possible gene pairs, with weak matches removed using the minbit criterion of 0.5. The *anvi-pan-genome* program uses a Markov Cluster Algorithm to group genes into putatively homologous groups called "gene clusters." We set the mcl-inflation parameter to 10 as suggested by Anvi'o for comparing very closely related genomes (https://merenlab.org/2016/11/08/pangenomics-v2/). Amino acid sequences within gene clusters were aligned with MUSCLE (v3.8.1551) (30). Finally, we performed hierarchical clustering across gene clusters and genomes using Euclidean distance and Ward linkage. This resulted in a pangenome showing the distribution of core and accessory genes across the reference genomes.

### Phylogeny, ANI, and comparison with GTDB

We constructed a phylogenetic tree based on the amino acid sequences of 71 bacterial single-copy core genes (24, 25). We first used the Anvi'o program *anvi-get-sequences-for-hmm-hits* to align protein sequences using MUSCLE (v3.8.1551) (30), concatenate gene sequences, return only the most significant hit, and output amino acid sequences. Only genes that occurred in at least 50% of the genomes were used for the analysis, which

in this case included all 71 genes. We trimmed alignments using trimAl (31) with the setting "-gt 0.5" to remove all positions that were gaps in more than 50% of sequences. Maximum likelihood phylogenetic trees were then computed using IQ-TREE (32) with the WAG model (33) and 1000 bootstrap replicate support. We included a type strain genome for *Escherichia coli* (strain ATCC 11775; GCA_003697165.2) to root the trees. To estimate pairwise whole genome ANI between the selected reference genomes in the pangenome, we used the Anvi'o program *anvi-compute-genome-similarity* with the parameters "--program pyANI" and "--method ANIb." To compare genomes against the classification in GTDB, we used GTDB-Tk (version 2.3.0) (34) with classify_wf and the R214 reference data release.

## Distribution of *Haemophilus* and *Aggregatibacter* genomes across human oral sites

We analyzed the distribution of natural populations of *Haemophilus* and *Aggregatibacter* species across human oral sites by mapping shotgun metagenomic sequences from the National Institutes of Health Human Microbiome Project (HMP) (35, 36) to our curated set of genomes. To obtain data from the HMP portal (https://portal.hmpdacc.org/), we searched for metagenomes using the following parameters: oral sites (buccal mucosa, supragingival plaque, subgingival plaque, dorsum of tongue, hard palate, palatine tonsil, throat, and saliva), Healthy Human Study (HHS), fastq files (FASTQ), and whole genome sequencing (wgs_raw_seq_set). The metagenomes consisted of ~100 bp paired-end reads that were sequenced from samples collected from nine oral sites in phases I and II of the HMP.

We performed quality filtering using *iu-filter-quality-minoche*,(37) which is based on recommendations from Minoche, Dohm, and Himmelbauer (38) for Illumina sequencing data. This resulted in a total of 2.5 billion quality-filtered metagenomic short reads from 686 samples across nine different oral sites, including three main sites in the oral cavity with large sample sizes (the buccal mucosa ($n = 183$), supragingival plaque ($n = 210$), and tongue dorsum (TD) ($n = 220$)) and six other sites with smaller sample sizes (subgingival plaque ($n = 19$), keratinized gingiva ($n = 14$), hard palate ($n = 1$), palatine tonsil ($n = 19$), throat ($n = 13$), and saliva ($n = 7$)).

We competitively mapped individual quality-filtered metagenomic samples to a concatenated file of the selected reference genomes in the pangenome using bowtie2 v2.4.1 (39) with the "--very-sensitive," "--end-to-end," and "--no-unal" flags. Competitive mapping assigns each mapped read to a single genome that provides the best match. BAM files were generated from the read alignment data using Samtools v1.9 (40), and the Anvi'o program *anvi-single-profile* was used to create a profile database containing coverage data for each metagenome. Profiles were then merged for each oral site using *anvi-merge*. We then extracted the mean depth of coverage and breadth of coverage for reads aligned to each genome using *anvi-summarize*. We classified a genome as detected in a metagenomic sample when the breadth of coverage was at least 50%. We then calculated the relative abundance of each detected genome by averaging its depth of coverage across nucleotide positions in which coverage was within the interquartile range (Q2Q3) and dividing by the total mean depth of Q2Q3 coverage for all reference genomes. We use the Q2Q3 quartiles of the mean depth of coverage to filter out outliers in coverage caused by mobile elements or other highly similar sequences shared among different taxa in the community. Only metagenomes in which at least one reference genome was detected were included in the calculation of relative abundance.

## Functional analysis of detected *Haemophilus* and *Aggregatibacter* genomes across human oral sites

We analyzed the distribution of functional annotations from the NCBI COG, KEGG, and Pfam databases across genomes detected at various oral sites. We started by predicting the metabolic capabilities of each reference genome using the Anvi'o script *anvi-estimate-metabolism* (with parameters: --kegg-output-modes modules). This function

determines the enzymes present in a genome using KEGG Orthologs (KOs). We used the default module completion threshold of 0.75, which scores a metabolic pathway (module) as "complete" within a genome when at least 75% of the KOs in a module are present. We then used the Anvi'o script *anvi-compute-metabolic-enrichment* to identify complete modules significantly enriched in one set of genomes compared to another. This script uses a generalized linear model with a logit link function to obtain enrichment scores and adjusted q-values for each combination of pairwise group comparisons. We also used the Anvi'o script *anvi-compute-functional-enrichment* to identify significantly enriched functional annotations from NCBI COG, KEGG, and Pfam, independent of metabolic pathway completeness. To determine whether the identified genes of interest were represented in oral metagenomes as well as in the reference genomes, we used *anvi-get-split-coverages* and a custom R script adapted from Utter et al. (9) to inspect the nucleotide coverage of these genes.

## RESULTS

### Oral *Haemophilus* and *Aggregatibacter* pangenome

The pangenome (Fig. 1) organized genomes based on hierarchic clustering of shared gene content. This clustering produced groups substantially concordant with the taxonomic names assigned by NCBI (illustrated in Fig. 1 through the color-coded genome layers), as well as the names assigned to genomes in the Genome Taxonomy Database (GTDB) (see Table S2), and the species of *Haemophilus* and *Aggregatibacter* recognized in the human oral cavity by HOMD. However, disparities were also evident. One major group contained the entire *Aggregatibacter* genus plus *Haemophilus*

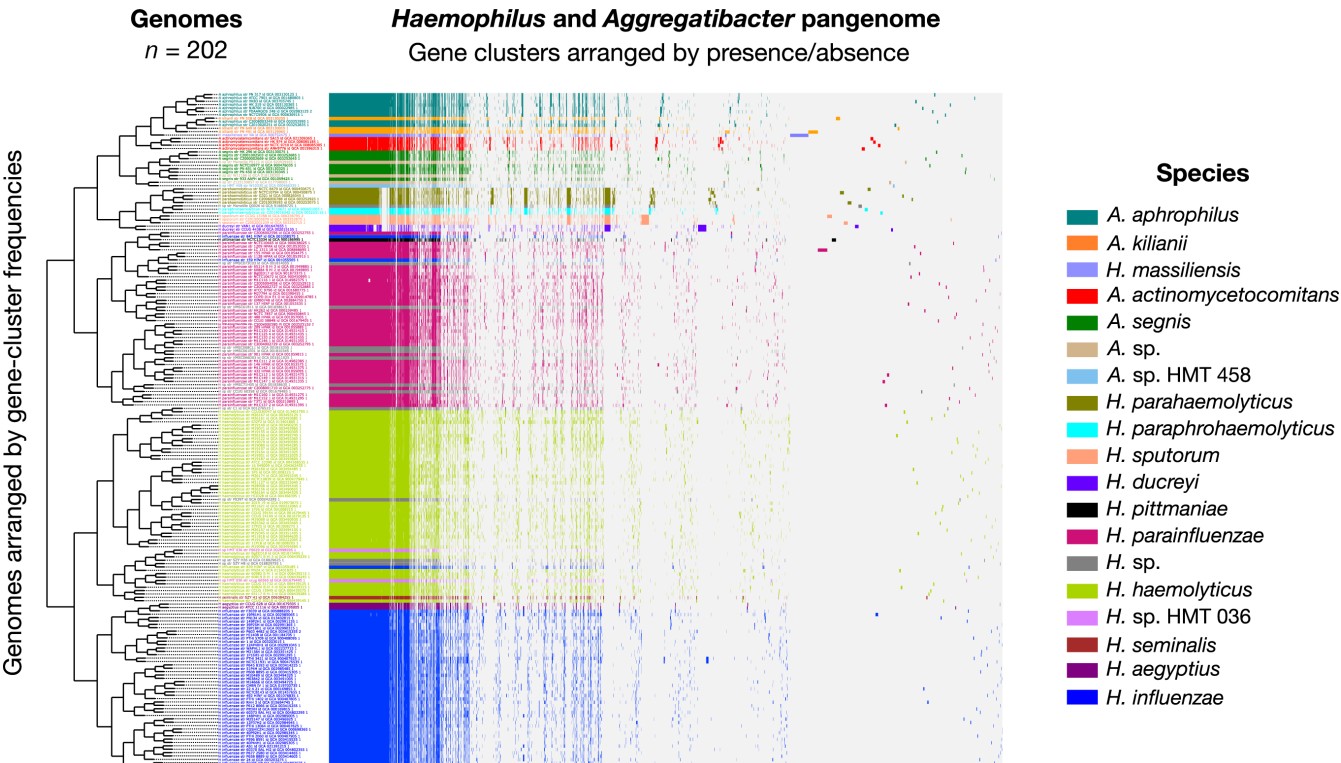

**FIG 1** Human oral *Haemophilus* and *Aggregatibacter* species pangenome constructed from (*n* = 202) NCBI RefSeq genomes. For all genomes, open reading frames (ORFs) were predicted, and NCBI blastp was used to calculate amino acid sequence similarity between all possible gene pairs, and a Markov Cluster Algorithm was used to cluster similar sequences to identify homologous genes (i.e., gene clusters). Gene clusters are colored by species and arranged based on their presence or absence across the genomes. Genomes are hierarchically clustered based on gene cluster frequency (i.e., the number of representatives of each gene cluster present in each genome), which is shown by the dendrogram on the left. This pangenomic analysis results in distinct groups by species and predicts the species identities of the unnamed *Haemophilus* genomes (*H.* sp.; colored gray) and *Aggregatibacter* genomes (*A.* sp.; colored tan).

*parahaemolyticus*, *Haemophilus paraphrohaemolyticus*, *Haemophilus sputorum,* and *Haemophilus ducreyi*. The *H. parainfluenzae* group included two genomes potentially misclassified as *H. influenzae*, one *Haemophilus pittmaniae* genome, and seven *Haemophilus* genomes identified in the NCBI only to the genus level as *H.* sp. Another group was composed primarily of *H. haemolyticus* genomes but also included a subgroup of *H. seminalis*, *H.* sp. HMT-036, one genome misclassified as *H. influenzae* and four *Haemophilus* genomes identified in the NCBI only to genus level. Genomes identified as *H. influenzae* clustered together.

Phylogenetic analysis based on single-copy core genes (SCG) was broadly concordant with the pangenomic groupings, although the arrangement of some genomes within specific groups differed slightly between the two methods of genomic organization (Fig. 2A). One exception of a single *H. influenzae* genome was notable, where it appears differently positioned in the phylogenomic tree compared to the pangenomic tree. Upon examining the concatenated amino acid sequences for this genome, we discovered that 69 out of the 71 extracted amino acid sequences presented an unusually high number of gaps, leading to its unusual placement between *H. parainfluenzae* and *H. haemolyticus*. This observation underscores the complexities and limitations of relying solely on phylogenomic approaches for determining genomic relationships and highlights the need for incorporating complementary methods, such as pangenomic analyses and average nucleotide identity assessments. Phylogenetic analysis was complemented by pairwise comparisons of genomic sequences using ANI. As depicted in Fig. 2B (also see Table S3), the overall ANI clustering pattern was congruent with both pangenomic and phylogenomic grouping. A threshold of 95% ANI appropriately delineated most

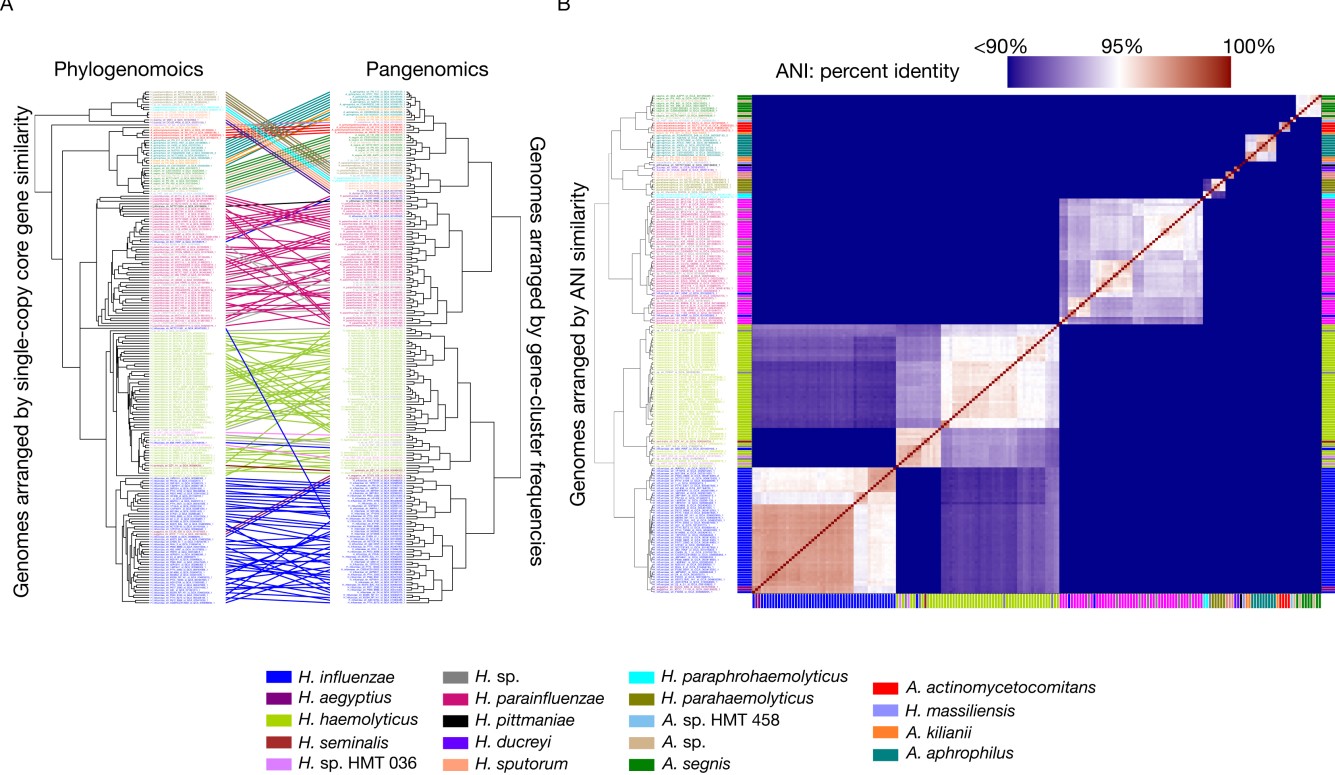

**FIG 2**   (A) Phylogenomic and pangenomic tree comparisons of *Haemophilus* and *Aggregatibacter* reference genomes cluster genomes into the same species-level groups. Rectangle color indicates species. The phylogenomic tree was constructed using maximum likelihood with concatenated single-copy core genes. The pangenomic tree was constructed using the gene frequencies present in each genome. Lines connect rectangles that represent the same genome. (B) Average nucleotide identity (ANI) comparison of *Haemophilus* and *Aggregatibacter* reference genomes included in the pangenome. ANI represents genome-level similarity at the nucleotide level between any two genomes and reveals distinct species, corroborating the results of pangenome clustering. Rectangle color indicates species, as in panel A. The color scale denotes genome percent similarity: 100% is red, 95% is white, and 90% and below is blue.

species boundaries, but disparities were also evident. For example, many *H. parainfluenzae* genomes had intraspecies ANI values below the 95% ANI threshold, suggesting the existence of distinct sub-species groups. Although no one measure of genomic similarity is without limitation, the three metrics taken together serve to provide a foundation for understanding the genomic relationships within the *Haemophilus* and *Aggregatibacter* genera.

## Distribution of *Haemophilus* and *Aggregatibacter* genomes across human oral sites

Distinct taxa often exhibit varying ecological preferences and distributions within the human oral ecosystem. The presence or absence of sets of shared gene clusters characterizes these taxonomic groupings, and by mapping metagenomic reads, we can evaluate the distribution of these groups across sampled oral habitats. Pangenomic groups displayed markedly different distributions among oral sites (Fig. 3). Notably, *Aggregatibacter* genomes were detected solely or primarily in dental plaque, including *A. aphrophilus*, *A.* sp. HMT-458, *A. segnis,* and *Aggregatibacter kilianii*. The two genomes classified as *A.* sp. HMT-458 were detected in nearly 60% of supragingival plaque samples, highlighting the importance of this unnamed species of *Aggregatibacter*. One *A. kiliani* genome (strain PN491) and a *Haemophilus massiliensis* genome that clustered with *Aggregatibacter* genomes based on gene content, yet were highly divergent from all other genomes based on ANI and phylogeny, were undetected in any samples. Similarly, the periodontal disease-associated *A. actinomycetemcomitans* was also undetected in any sample, which was expected given that our samples were obtained from healthy subjects.

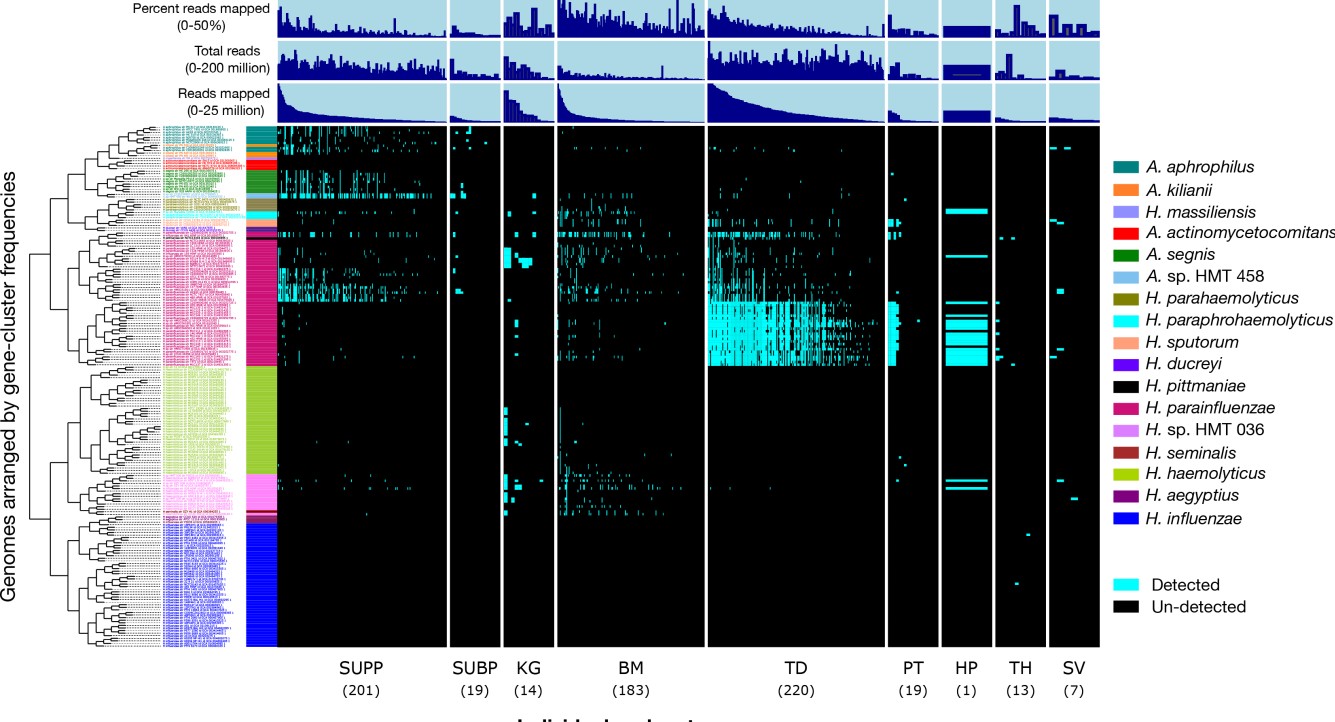

**FIG 3** Detection plot of *Haemophilus* and *Aggregatibacter* species and strains in 686 Human Microbiome Project (HMP) metagenomic samples from nine major oral sites. Each row displays the detection of a genome across all samples. A genome is detected (cyan) if at least 50% of its nucleotides have at least 1X coverage. If the genome is not detected in a sample, it is represented by a black bar. Samples are ordered by oral site and then by the decreasing number of reads mapped to the set of genomes. From left to right, oral sites are supragingival plaque (SUPP; *n* = 210), subgingival plaque (SUBP; *n* = 19), keratinized gingiva (KG; *n* = 14), (TD; *n* = 220 samples), palatine tonsil (PT; *n* = 19), throat (TH; *n* = 13), saliva (SV; *n* = 7), hard palate (HP; *n* = 1), and buccal mucosa (BM; *n* = 183). Additional data are shown for "total reads" in each sample, the number of "reads mapped" per sample, and "percent reads mapped" per sample.

The landscape of the human oral cavity harbors diverse habitats that may enable species- or strain-level habitat specializations. Patterns of site-specialization were particularly evident for *H. parainfluenzae*, which, while generally prevalent across all oral sites, had pangenomic subgroups with clear habitat preferences (Fig. 3). One *H. parainfluenzae* subgroup was detected in 89% of TD samples and was nearly absent from other major oral sites, such as dental plaque and buccal mucosa. A second *H. parainfluenzae* subgroup was primarily detected in supragingival plaque samples and a third subgroup in keratinized gingiva. Conversely, two *H. parainfluenzae* genomes, one of which is potentially misclassified in NCBI as *H. influenzae*, were detected throughout most oral sites, suggesting a fourth subgroup of habitat generalists. This pattern of *H. parainfluenzae* subgroup habitat specialization is further supported by the inspection of the whole genome mean depth of coverage (Fig. S1). These findings are consistent with a prior study that demonstrated similar patterns of habitat specialization for *H. parainfluenzae* within the oral microbiome (9).

Another apparent habitat specialist is the *H.* sp. HMT-036 group, which was detected in 44% of buccal mucosa samples and 36% of keratinized gingiva samples, but less than 10% of samples from other habitats. The closely related *H. haemolyticus* group had a similar distribution to that of the *H.* sp. HMT-036 group but was much less prevalent, detected in less than 4% of samples from buccal mucosa. The putative pathogens *H. influenzae*, *H. aegyptius,* and *H. ducreyi* were not detected in any samples, with the exception of *H. influenzae,* which was detected in two throat samples.

While the detection of specific microbial genomes provides insights into presence or absence patterns, the relative abundance metric identifies dominant versus low-abundance taxa and provides additional information about their distribution within microbial communities. For example, the relative abundance results, as depicted in Fig. 4, show that a single subgroup of *H. parainfluenzae* dominates the TD, where it constituted over 50% of the *Haemophilus* or *Aggregatibacter* taxa abundance in more than 90% of the samples and was largely absent or at low abundance in dental plaque or buccal mucosa. Except for *A. kilianii*, which exhibits a distinct distribution pattern, other taxa that are highly abundant in supragingival plaque, such as *A. aphrophilus*, also tend to show high abundance in subgingival plaque, without any clear distinction in the distribution patterns between strains. This observation suggests adaptation of these taxa to environments that are common to both supra- and sub-gingival plaques. In many metagenomic samples, an interesting phenomenon emerges—the dominance of a single *Haemophilus* or *Aggregatibacter* species. For example, keratinized gingiva contains either *H.* sp. HMT-036 or a subgroup of *H. parainfluenzae* as the most prevalent taxon. In supragingival plaque, there appears to be a three-way reciprocal relationship among *H. parainfluenzae*, A. sp. HMT-458, and *A. aphrophilus* (Fig. S2), in which only one of the three taxa is in high abundance in any given sample. The significance of this reciprocal three-way relationship is supported through a permutation test (*P*-value = 0.0384; 95% CI = (0.0353, 0.0414); see Supplemental Materials: *Permutation test for three-way reciprocal relationship*).

To comprehensively understand microbial communities, it is critical to not only consider genome-level metrics but also examine individual gene-level coverage. This allows us to assess the array of genes that are exclusive to the isolates and not found in the environment, as well as iscover genes that are abundant in one specific habitat but scarce or absent in others. We focused on two isolates with habitat tropisms at the genome level: *A.* sp. HMT-458 in supragingival plaque and *H. parainfluenzae* strain M1C142-1 in the TD. We classified a gene as detected in a sample if at least 90% of its nucleotides had at least 1X coverage. Using a threshold of 90%, rather than 100%, allowed for the detection of genes that differed slightly between reference genomes and environmental genomes. We then created a binary gene detection map for the 30 metagenomic samples with the greatest median coverage for each of the three main oral sites (supragingival plaque, TD, and buccal mucosa). Fig. 5 shows that for these two genomes a greater proportion of genes were detected within only one of

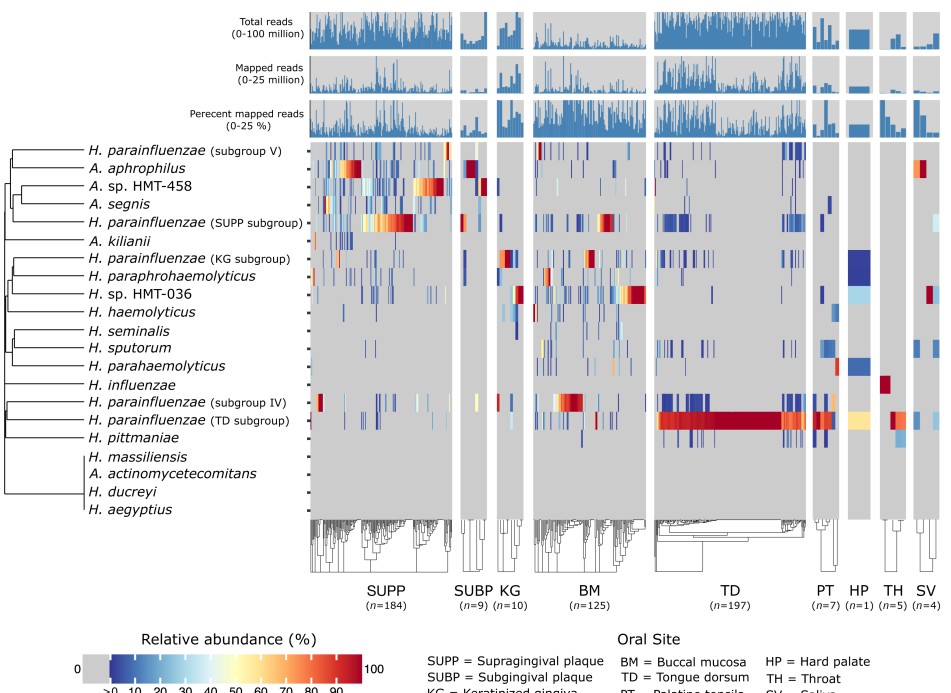

**FIG 4** Heatmap shows the relative abundance of *Haemophilus* and *Aggregatibacter* species concerning the set of reference genomes across nine major oral sites: supragingival plaque (SUPP), subgingival plaque (SUBP), keratinized gingiva (KG), tongue dorsum (TD), palatine tonsils (PT), throat (TH), saliva (SV), hard palate (HP), and buccal mucosa (BM). Relative abundance for each species represents the sum of relative abundances of each reference genome from that species, calculated from the mean depth of coverage across nucleotide positions in the 2nd and 3rd quartiles (the Q2Q3 interquartile range) after nucleotides were ranked by their depth of coverage and then divided by the sum of mean coverages of all genomes within a metagenomic sample. The rows and columns correspond to individual species and Human Microbiome Project metagenomic samples, respectively. Only metagenomes in which at least one reference genome was detected (i.e., at least 50% of its nucleotides have at least 1X coverage) were included. The number of samples for each oral site at which a species was detected is listed in the figure. For clearer visualization, we inflated the width of oral sites with small sample sizes. Species (y-axis) and samples within oral sites (x-axis) are hierarchically clustered based on Bray–Curtis distances.

the three major oral habitat types of dental plaque, TD, and buccal mucosa. For *A*. sp. HMT-458, the proportion of genes detected in supragingival plaque samples was nearly six-fold greater than those of TD and buccal mucosa. Similarly, for *H. parainfluenzae* strain M1C142-1, a nearly five-fold higher proportion of genes was detected in TD samples compared to supragingival plaque or buccal mucosa. For a third reference genome, *H. parainfluenzae* strain NCTC-7857, gene detection was distributed more evenly across oral sites, indicating a more generalized distribution. Gene detection maps also revealed a small proportion of genes that attracted metagenomic reads from multiple habitats. Many of these genes were annotated with highly conserved functions or annotated as transposable elements. Read recruitment to such genes may represent cross-mapping to different taxa that share high similarity in these genes or share elements that have been horizontally acquired. Overall, gene-level coverage analysis confirmed the habitat tropisms suggested by whole genome-level coverage analysis and extended the analysis by identifying individual genes present in specific oral habitats.

## Functional analysis of *H. parainfluenzae* genomes across human oral sites

To identify functions that might drive the distribution of *H. parainfluenzae* subgroups to TD or supragingival plaque, we conducted a gene functional enrichment analysis, focusing particularly on the TD and supragingival plaque subgroups because they demonstrated sufficient prevalence to be analyzed in their respective environmental

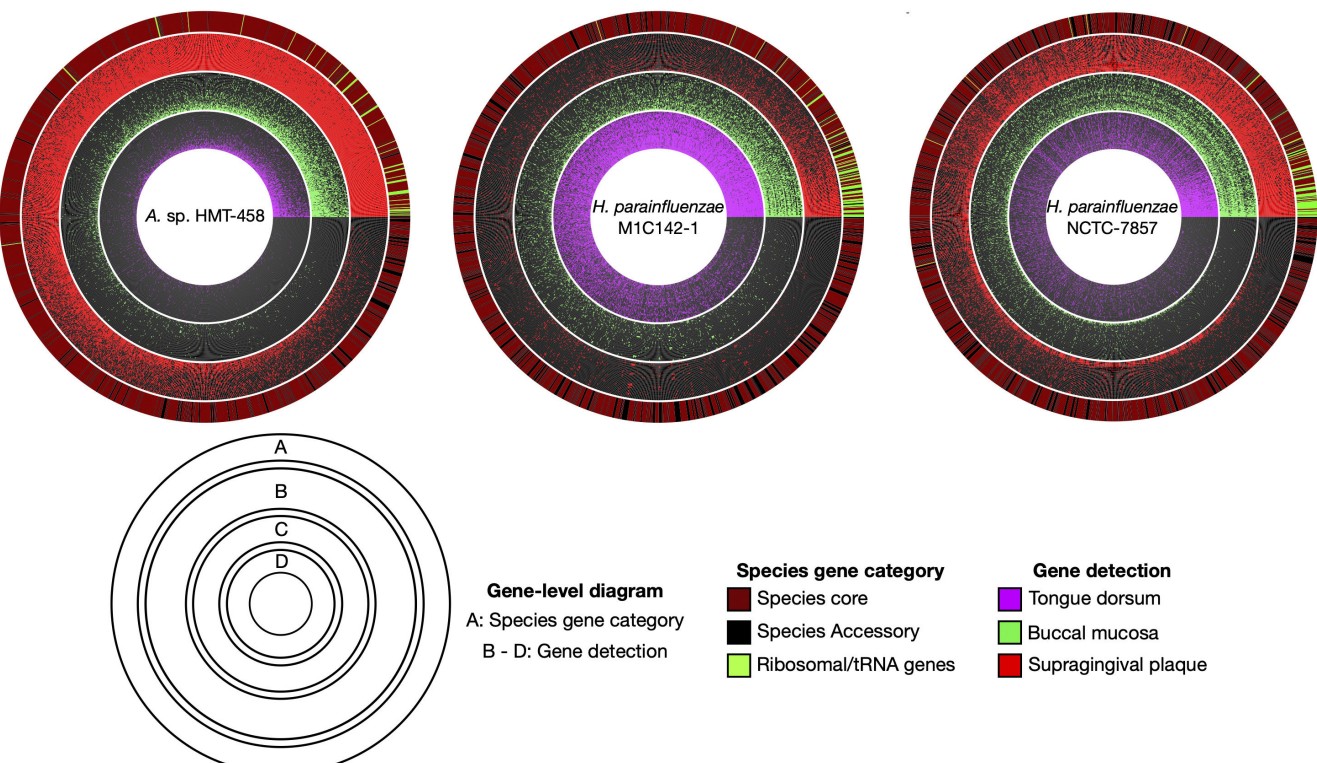

**FIG 5** Gene-level detection diagrams for select *Haemophilus* and *Aggregatibacter* genomes illustrate patterns of site-specialization. Here, we display radial gene-level detection maps for three genomes showing the detection of genes across a subset of the top 30 metagenomic samples ranked by median coverage from three oral sites (supragingival plaque (red), buccal mucosa (green), and tongue dorsum (magenta)). *A.* sp. HMT-458 and *H. parainfluenzae* strain M1C142-1 represent the most abundant species based on mean coverage within their respective preferred oral sites. *H. parainfluenzae* strain NCTC-7857 illustrates a more generalized distribution pattern in which the proportion of genes detected is relatively even across oral sites. A gene was classified as detected within an oral site (colored red, green, or magenta, respectively) when at least 90% of the nucleotides of the gene had at least 1X coverage. Genes are ordered according to their detection frequencies, and samples within each oral site are ordered from inward to outward based on the proportion of genes detected. The outermost layer for each diagram indicates whether a gene is core (brick red) or accessory (black) for each respective species. A group of ribosomal genes translated using Hidden Markov Models through the Anvi'o program anvi-run-hmms are indicated in bright green.

metagenomes (Fig. 4; Table S4). This analysis showed that both the tongue-resident and the dental plaque-resident *H. parainfluenzae* had unique pathways. The *H. parainfluenzae* SUPP subgroup had significantly enriched metabolic pathways for biotin biosynthesis (KEGG module ID M00577, M00123, and M00572), histidine biosynthesis (KEGG module ID M00026), and a thiamine salvage pathway (KEGG module ID M00899). These pathways were consistently incomplete in the TD subgroup (Table S5). Further functional enrichment analysis using data from the NCBI COG20, Pfam, and KEGG databases supported these findings and revealed that the *BioA* and *BioF* genes were exclusive to the *H. parainfluenzae* SUPP subgroup. In contrast, the completeness of the thiamine salvage and histidine biosynthesis pathways varied, with a small fraction of the *H. parainfluenzae* TD subgroup having complete pathways.

The *H. parainfluenzae* TD subgroup exclusively possessed three genes from the biotin-dependent Oad gene operon (OadA1, OadB, and OadG), corroborating a previous study (9). These genes encode the three subunits of a sodium-dependent oxaloacetate decarboxylase enzyme, which performs the conversion of oxaloacetate to pyruvate while simultaneously translocating two sodium ions from the cytoplasm to the periplasm, providing a shunt to gluconeogenesis and establishing a potentially useful Na +gradient (41, 42). In contrast, no complete oxaloacetate decarboxylase operon was detected in any of the other *H. parainfluenzae* genomes, and no additional functions were exclusively present in the TD-associated subgroup.

To determine whether these functions, characteristic of the cultivated isolates of these subgroups, were also characteristic of the community in the oral habitats, we inspected the average coverage of each gene in the three most prevalent oral sites (TD, supragingival plaque, and buccal mucosa). All three Oad genes exhibited substantially greater coverage in TD samples in comparison to supragingival plaque and buccal mucosa samples. Moreover, the coverage per sample corresponded with the *H. parainfluenzae* species-level coverage averaged across all 54 *H. parainfluenzae* genomes (Fig. S3 to S5). Similarly, the coverage of biotin biosynthesis genes (*BioA* and *BioF*) enriched in the supragingival plaque subgroup was markedly low in TD samples compared to supragingival plaque and buccal mucosa samples, averaging less than 1X depth of coverage at the species level (Fig. S6 and S7). The differential coverage patterns of these specific genes across various oral habitats underscore the niche-specific genetic adaptations and metabolic capabilities of *H. parainfluenzae* subgroups within the human oral cavity.

## Functional analysis of the *H.* sp. HMT-036 group

To identify functions that could differentiate HMT-036 genomes from *H. haemolyticus* genomes, we conducted a gene functional enrichment analysis. The HMT-036 group exhibited significant enrichment in metabolic pathways for heme biosynthesis (KEGG module ID M00926, M00121) and siroheme biosynthesis (KEGG module ID M00846) compared to the *H. haemolyticus* genomes (Table S7). Every HMT-036 genome possessed a complete heme biosynthesis pathway, incorporating the *HemABCDEGHKNL* genes essential for heme synthesis from L-glutamate (43). Additionally, these genomes possess the *CysG* gene encoding uroporphyrin-III C-methyltransferase, which, when combined with the *HemBCD* genes, facilitates the creation of siroheme, a key component for both nitrite and sulfite reductases. In contrast, the thiamine salvage pathway (KEGG module ID M00899) was prevalent in *H. haemolyticus* genomes (28 of 42 genomes possessing a full pathway) but was absent in HMT-036 genomes. Furthermore, our research identified seven additional COG20 functions present across all HMT-036 genomes but missing in *H. haemolyticus* (Table S6). These include a peroxiredoxin (*Tpx*), 6-pyruvoyl-tetrahydropterin synthase (*QueD*), an adenosine deaminase (*Add*), soluble cytochrome b562 (*CybC*), sucrose-6-phosphate hydrolase (*SacC*), an ABC-type transport enzyme (*YpjD*), and an undefined membrane protein (*YqgA*). It is worth noting that all *Haemophilus* and *Aggregatibacter* genomes in our pangenomic study contained at least one gene encoding for peroxiredoxin, adenosine deaminase, and several ABC-type transporters. However, cytochrome b562 appeared only in a minor group of *Haemophilus* species, while sucrose-6-phosphate hydrolase was distinctly absent in the *H. haemolyticus* and *H. influenzae* genomes. This comparative genomic analysis underscores the distinct metabolic capabilities of HMT-036, potentially reflecting evolutionary adaptations that distinguish it from *H. haemolyticus* and offering insights into its unique biological functions and ecological niche.

## DISCUSSION

Metapangenomic analysis of *Haemophilus* and *Aggregatibacter* genera revealed habitat specialization, metabolic versatility, and insights into taxonomic classifications. Notably, our data support that *Aggregatibacter* species predominantly inhabit dental plaque. In contrast, an as-yet-unnamed *Haemophilus* species, designated HMT-036, shows an affinity for the buccal mucosa and keratinized gingiva. Furthermore, strains within *H. parainfluenzae* appear to be uniquely adapted to different oral habitats. These findings bolster the site-specialist hypothesis of the human oral microbiome, which posits that distinct regions within the mouth support profoundly different microbial communities (7). Historically, the foundation for this hypothesis has largely rested on 16S rRNA gene sequence data (8, 44, 45). However, this method often cannot distinguish between closely related species and strains. As such, genome-wide analysis is indispensable when trying to identify habitat specialization in complex natural microbial communities that

consist of closely related taxa. By harnessing the enhanced clarity offered by whole-genome sequencing data, we determined the preferred habitats of *Aggregatibacter* and *Haemophilus* species, further strengthening the validity of the site-specialist hypothesis.

The potential three-way reciprocal relationship among closely related species, *H. parainfluenzae*, *A. aphrophilus*, and *A.* sp. HMT-458, where one of the three dominates in each supragingival plaque sample, is an interesting pattern. This pattern may result from microbial warfare, in which bacteria secrete compounds such as bacteriocins that inhibit the growth of competitors. We are unaware of any study of the production of bacteriocins by *H. parainfluenzae*, *A. aphrophilus*, or *A.* sp. HMT-458, but there is strong evidence that the production of bacteriocins by *H. influenzae* and *H. haemolyticus* plays an important role in their colonization in host tissues through direct negative effects on other competing species (46, 47). An alternative explanation for the dominance of one of the three species in each sample may be that microenvironments are heterogeneously distributed throughout dental plaque, and these three species might exhibit preferences for different microenvironments. For example, recent evidence from spectral imaging fluorescence *in situ* hybridization shows that *H. parainfluenzae* is distributed in loose clusters adjacent to streptococci in coccus-rich plaque (48), whereas a different member of the Pasteurellaceae, possibly *A.* sp. HMT-458, is present in multi-species "Corncob" structures in filament-rich "hedgehog" structures of plaque (49, 50). Thus, the presence or absence of corncob and hedgehog structures in a dental plaque sample may explain the concomitant presence or absence of *A.* sp. HMT-458. The dominance of a single species at a given time may also be due to temporal fluctuations in growth rates or environmental conditions that favor one species over the others, and the observed dominance might be a snapshot of an ongoing microbial succession. Finally, differences between human subjects in their oral hygiene practices, diet, or genetics may influence environmental factors, such as pH, oxygen levels, nutrient availability, and temperature, that could favor the growth of one species over the others.

One important factor in determining bacterial colonization is the availability of micronutrients, such as inorganic cofactors (e.g., iron) and coenzymes, that maintain normal metabolism. The human oral cavity has an exceptionally limited supply of free iron, with most of the iron bound to host-produced proteins (51). In response, *Haemophilus* and *Aggregatibacter* species have evolved a variety of ways to acquire iron from the host environment, including the secretion of siderophores that solubilize and bind to an external source of iron with high affinity, as well as the production of iron–porphyrin heme that can later be broken down to release iron (52–54). Our mapping data showed that *Haemophilus* species that lack the ability to synthesize heme, including *H. influenzae*, *H. aegyptius,* and *H. haemolyticus*, were undetected or consistently rare in the healthy human oral microbiome. Conversely, *Haemophilus* and *Aggregatibacter* species capable of independent heme biosynthesis were highly abundant throughout the human oral cavity. These results agree with a recent comparative genomics study proposing that the shared ancestor of present-day *H. haemolyticus* and *H. influenzae* lost heme biosynthesis genes during their evolution away from *H. parainfluenzae,* possibly due to adapting to specific ear, nose, and throat environments (15). This pattern complements a growing body of literature that highlights the pivotal role of micronutrients in shaping microbial community structures (4, 55–57).

The ability of an organism to metabolize a variety of substrates as energy sources gives it a competitive advantage in diverse habitats or under fluctuating environmental conditions. Metabolically versatile organisms possess the enzymatic machinery to shift between different metabolic pathways. Central to bacterial metabolic versatility is their ability to toggle between glycolysis, a catabolic pathway breaking down glucose to yield ATP, and gluconeogenesis, its anabolic counterpart synthesizing glucose from non-carbohydrate precursors, especially during fasting or low carbohydrate intake (58). A key link between glycolysis, gluconeogenesis, and the tricarboxylic acid (TCA) cycle is the PEP–pyruvate–oxaloacetate node, a crucial switch point directing the carbon flux in response to environmental conditions (59). In TD samples, we previously discovered

a unique sub-species group of *H. parainfluenzae* equipped with genes for oxaloacetate decarboxylase (OAD), which was absent in their counterparts from supragingival plaque. This enzyme plays a significant role in converting oxaloacetate, an intermediary in the TCA cycle, to pyruvate (60), a precursor for several metabolic pathways. Such capability suggests that this *H. parainfluenzae* subgroup can efficiently toggle between metabolic pathways, adapting to the variable nutrient availability on the tongue. Conversely, in the relatively stable environment of the supragingival plaque, constantly exposed to dietary sugars, such enzymatic flexibility might be less critical. Additionally, OAD's role in sodium ion transport (60) might serve a dual purpose, facilitating energy production and maintaining intracellular pH under the variable conditions on the tongue. Overall, the presence of this enzyme in *H. parainfluenzae* tongue specialists illuminates the sophisticated evolutionary adaptations that allow bacteria to specialize and thrive in the distinct niches of the oral ecosystem.

Another important factor that may influence bacterial ecology is the ability to synthesize vitamins that are difficult to obtain in a habitat. Our findings indicate that *H. parainfluenzae* tongue specialists lack the ability to synthesize the vitamin biotin *de novo*. Biotin is an enzyme cofactor that is necessary for the survival of all living organisms, including bacteria (61). According to the Kyoto Encyclopedia of Genes and Genomes (KEGG; www.kegg.jp), there are five pathway modules associated with biotin biosynthesis, where each module represents a series of reactions to produce specific end-products. All *Haemophilus* and *Aggregatibacter* genomes in our data set are missing the genes for the enzymes *BioI*, *BioU,* and *BioW*, effectively ruling out three pathways that lead to biotin biosynthesis. This leaves two non-mutually exclusive pathways; the M00572 pathway that converts pimelic acid to pimeloyl-ACP catalyzed by *BioC* and *BioG*, and the M00123 pathway that converts pimeloyl-ACP-CoA to biotin catalyzed by *BioA*, *BioB*, *BioD*, *BioF,* and *BioG*. The *H. parainfluenzae* TD specialists lack several key genes in both pathways, including *BioA*, *BioC*, *BioF,* and *BioG*, indicating their inability to synthesize biotin *de novo*. In our data set, all other *Haemophilus* and *Aggregatibacter* genomes have the required genes to synthesize biotin, suggesting the loss of biotin biosynthesis genes in this group of *H. parainfluenzae*, likely due to specialized adaptation to the TD. The evolutionary loss of costly genes required to produce biotin is expected where an organism could obtain biotin or its precursors through other less costly means (62). Thus, it is likely that TD *H. parainfluenzae* specialists scavenge biotin or its precursors from the environment, potentially highlighting important interactions with other bacterial taxa. Further experimental research is needed to validate these findings.

Traditional classifications of human oral *Haemophilus* and *Aggregatibacter* species are challenged by recent genomic analysis, emphasizing the pivotal role of whole-genome sequencing in taxonomy. For example, our phylogenomics results suggest that *H.* sp. HMT-036 is a distinct lineage within the *H. influenzae/H. haemolyticus* clade. Pangenomics further revealed that the *H.* sp. HMT-036 genomes possess several functional traits that distinguish this species from *H. influenzae* and *H. haemolyticus*. Additionally, our mapping results indicate that *H.* sp. HMT-036 is a common member of the healthy oral microbiome, showing distinct distribution patterns, and is particularly prevalent in the soft tissues of keratinized gingiva and buccal mucosa. Past attempts to classify genomes closely related to *H. influenzae* and *H. haemolyticus* have faced challenges due to the high sequence similarity of conserved marker genes (63). However, a recent comparative genomics study of the *Haemophilus* genus revealed strains provisionally named "*H. intermedius*" group to be haemin-independent *H. haemolyticus* (15). Therefore, *H.* sp. HMT-036 might correspond to the unofficially named species "*H. intermedius*," first proposed in 1989 (10). Considering these findings, it is increasingly evident that genomic-scale analyses are an invaluable resource for taxonomy, refining our understanding of the nuanced differences between closely related species.

Metapangenomics can also reveal potential species misclassifications of reference genomes that can distort our interpretations of the genetic relationships, evolutionary histories, and functional traits of bacterial species. Our analysis revealed similarities in

gene content and distribution patterns of multiple reference genomes, indicating that they are potentially misclassified to species level in NCBI. For example, two genomes identified in NCBI as *H. influenzae* clustered with *H. parainfluenzae* within the pangenome and followed a similar distribution pattern in the human oral cavity as *H. parainfluenzae*. Phylogenomics and ANI further indicate that these reference genomes are significantly divergent from other *H. influenzae* genomes, strongly suggesting that they are misclassified. In addition to misclassification, for many years there has been a controversy about whether *H. aegyptius* should be classified separately from *H. influenzae*.(10) Our results based on phylogenomics and ANI support the argument that they do not merit separate species rank. Uncovering such misclassifications can lead to a more accurate and refined understanding of biodiversity, evolution, and ecological interactions.

## Conclusion

Metapangenomics can provide information on the ecological distribution and genetic variation among bacterial species and strains in natural habitats. Leveraging this methodology, we explored the ecological niche partitioning of *Haemophilus* and *Aggregatibacter* species across various habitats within the human oral cavity, establishing evidence of distinctive site-specialization patterns, such as *Aggregatibacter* species in dental plaque, a distinct subgroup of *H. parainfluenzae* on the TD, throat, and tonsils, and the recently discovered *H.* sp. HMT-036 on keratinized gingiva and buccal mucosa. Additionally, we found evidence of a tripartite reciprocal relationship among closely related taxa residing in dental plaque, with one species dominant in each sample. Beyond characterizing the habitat tropism of species, our method permits analysis of gene distribution and abundance across genomes and oral samples. Namely, the systematic analysis of unique or over-represented genes and functions within site-specific *H. parainfluenzae* genome groups identified genes associated with biotin biosynthesis and oxalacetate decarboxylase that may facilitate the adaptation of various *H. parainfluenzae* sub-species groups to their respective oral niches.

## ACKNOWLEDGMENTS

We thank A. Murat Eren and the Anvi'o team for their assistance, consultations, and support in the use of Anvi'o. We thank Rich Fox for expert systems administration and assistance with using the Josephine Bay Paul Center servers at the Marine Biological Laboratory.

This work was supported by NIH grants R01 DE030136, R01 DE022586, and 2R01 DE016937.

All authors participated in the design of the study. J.J.G. performed the analysis. J.J.G., G.G.B., and J.L.M.W. drafted the manuscript. All authors critically revised the manuscript and approved the final version.

## AUTHOR AFFILIATIONS

[1]The Forsyth Institute, Cambridge, Massachusetts, USA
[2]Harvard School of Dental Medicine, Boston, Massachusetts, USA
[3]Marine Biological Laboratory, Woods Hole, Massachusetts, USA

## AUTHOR ORCIDs

Jonathan J. Giacomini http://orcid.org/0000-0002-0151-894X
Floyd E. Dewhirst http://orcid.org/0000-0003-4427-7928

## FUNDING

| Funder | Grant(s) | Author(s) |
| --- | --- | --- |
| HHS | National Institutes of Health (NIH) | R01 DE030136 | Floyd E. Dewhirst |

| Funder | Grant(s) | Author(s) |
|---|---|---|
| | | Gary G. Borisy |
| | | Jessica L. Mark Welch |
| HHS \| National Institutes of Health (NIH) | R01 DE022586 | Floyd E. Dewhirst |
| | | Gary G. Borisy |
| | | Jessica L. Mark Welch |
| HHS \| National Institutes of Health (NIH) | 2R01 DE016937 | Floyd E. Dewhirst |
| | | Gary G. Borisy |
| | | Jessica L. Mark Welch |

## AUTHOR CONTRIBUTIONS

Jonathan J. Giacomini, Conceptualization, Data curation, Formal analysis, Investigation, Methodology, Project administration, Resources, Software, Supervision, Validation, Visualization, Writing – original draft, Writing – review and editing | Julian Torres-Morales, Conceptualization, Methodology, Writing – review and editing | Jonathan Tang, Conceptualization, Methodology, Writing – review and editing | Floyd E. Dewhirst, Conceptualization, Funding acquisition, Project administration, Writing – review and editing | Gary G. Borisy, Conceptualization, Funding acquisition, Project administration, Writing – original draft, Writing – review and editing.

## DATA AVAILABILITY

The raw data used in this study are publicly available at NIH GenBank and RefSeq (https://www.ncbi.nlm.nih.gov/genome/) for genomes and HMP metagenomes from https://portal.hmpdacc.org/. The code used for analyses is available on Github (https://github.com/FatherofEverest/Spatial-ecology-of-Haemophilus-and-Aggregatibacter-the-human-oral-cavity).

## ADDITIONAL FILES

The following material is available online.

### Supplemental Material

**Fig. S1 (Spectrum04017-23-S0001.pdf).** Genome mean coverage plot.
**Fig. S2 (Spectrum04017-23-S0002.pdf).** Supragingival plaque ternary plot of relative abundance between three dominant plaque species.
**Fig. S3 (Spectrum04017-23-S0003.pdf).** OAD-A1 gene, genome, and species coverage plot.
**Fig. S4 (Spectrum04017-23-S0004.pdf).** OAD-G gene, genome, and species coverage plot.
**Fig. S5 (Spectrum04017-23-S0005.pdf).** OAD-B gene, genome, and species coverage plot.
**Fig. S6 (Spectrum04017-23-S0006.pdf).** BIOF gene, genome, and species coverage plot.
**Fig. S7 (Spectrum04017-23-S0007.pdf).** BIOA gene, genome and species coverage plot.
**Fig. S8 (Spectrum04017-23-S0008.pdf).** Oad gene operon nucleotide level coverage plot.
**Supplemental material (Spectrum04017-23-S0009.docx).** Supplemental text and figure legends.
**Tables S1 to S8 (Spectrum04017-23-S0010.xlsx).** Additional experimental details and results referenced in the main text, including reference genome metadata, mapping statistics, functional enrichment results, Oad and Bio gene coverage data, and pairwise ANI comparisons.

Open Peer Review

**PEER REVIEW HISTORY (review-history.pdf).** An accounting of the reviewer comments and feedback.

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
