## [Reviewer comments · Microbiology Spectrum]

Microbiology Spectrum

Spatial ecology of *Haemophilus* and *Aggregatibacter* in the human oral cavity

Jonathan Giacomini, Julian Torres-Morales, Jonathan Tang, Floyd Dewhirst, Gary Borisy, and Jessica Mark Welch

Corresponding Author(s): Jonathan Giacomini, The Forsyth Institute

Review Timeline:

Submission Date:	November 23, 2023
Editorial Decision:	January 9, 2024
Revision Received:	January 18, 2024
Accepted:	January 26, 2024

Editor: Justin Kaspar

Reviewer(s): Disclosure of reviewer identity is with reference to reviewer comments included in decision letter(s). The following individuals involved in review of your submission have agreed to reveal their identity: Clifford J. Beall (Reviewer #1)

Transaction Report:

DOI: <https://doi.org/10.1128/spectrum.04017-23>

Re: Spectrum04017-23 (Spatial ecology of *Haemophilus* and *Aggregatibacter* in the human oral cavity)

Dear Dr. Jonathan J. Giacomini:

Thank you for the privilege of reviewing your work. Below you will find my comments, instructions from the Spectrum editorial office, and the reviewer comments.

Thank you for your submission to Spectrum. As you will see, the referenced manuscript was well received by two reviewers, with very minor comments. I am only returning to you so that you may double check and provide any updates/fixes to the few areas reviewers suggest. We look forward to receiving your resubmission.

Revision Guidelines

Sincerely,
Justin Kaspar
Editor
Microbiology Spectrum

Reviewer #1 (Comments for the Author):

I enjoyed reading this, it is a great example of expanding knowledge using publicly available data. It's well written and the conclusions are well supported.

I only had some minor comments. In figure 1, absence is misspelled in the heading. In figure 2A, at least in the review copy pdf, parts of the dendrograms and some of the genome names aren't showing.

Another thing I noticed in Fig 2A was there is one genome that is labeled *Haemophilus influenzae* and sits in the middle of that group in the pangenomics tree but is on a long branch in the phylogenomics tree ending up between *parainfluenzae* and *haemolyticus*. I can't see a genome name for that one, but it seems like it might be worth checking to see if there is some kind of error there.

Reviewer #2 (Comments for the Author):

This manuscript detailed the results of a pangenomic analysis of two common oral bacteria genera: *Haemophilus* and *Aggregatibacter*. The introduction was well written and clearly outlined the research problem, the limitations of previous work, and the overall approach that was used in this study. The materials and methods were well detailed, with specific information about programs, commands, and criteria. Throughout the paper, appropriate references/citations were included. I also appreciated that the SI tables were included, so you can easily search for your own genes and/or organisms of interest. The work presented in this paper can easily be utilized to build new research hypotheses and design wet-lab experiments or reanalyze and mine existing datasets. Overall, I found the paper quite interesting and expect it will be compelling to a wide audience-general microbial ecologists as well as those specifically interested in evolution, pathogenesis, and/or the oral microbiome. Thank you for the great read!

Specific Comments:

Line 291: Is there a reason you focused on these two isolates in particular? For instance, I would be interested in understanding detailed differences, if any, between the *A. aphrophilus* in SUPP v SUBP.

Line 524: Your Github link doesn't work for me-maybe it's still private? I can find the user, but nothing listed in the profile that looks like it's associated with this work.

Review:

This manuscript detailed the results of a pangenomic analysis of two common oral bacteria genera: *Haemophilus* and *Aggregatibacter*. The introduction was well written and clearly outlined the research problem, the limitations of previous work, and the overall approach that was used in this study. The materials and methods were well detailed, with specific information about programs, commands, and criteria. Throughout the paper, appropriate references/citations were included. I also appreciated that the SI tables were included, so you can easily search for your own genes and/or organisms of interest. The work presented in this paper can easily be utilized to build new research hypotheses and design wet-lab experiments or reanalyze and mine existing datasets. Overall, I found the paper quite interesting and expect it will be compelling to a wide audience—general microbial ecologists as well as those specifically interested in evolution, pathogenesis, and/or the oral microbiome. Thank you for the great read!

Specific Comments:

Line 291: Is there a reason you focused on these two isolates in particular? For instance, I would be interested in understanding detailed differences, if any, between the *A. aphrophilus* in SUPP v SUBP.

Line 524: Your Github link doesn't work for me—maybe it's still private? I can find the user, but nothing listed in the profile that looks like it's associated with this work.

We are grateful to the reviewers for their thorough review and positive feedback on our manuscript. Reviewer 1 commended the work as "a great example of expanding knowledge using publicly available data" and appreciated its well-supported conclusions. Reviewer 2 found the paper "quite interesting" and relevant to a broad range of audiences, including microbial ecologists and those interested in evolution, pathogenesis, and the oral microbiome.

In response to the thoughtful comments provided by the reviewers, we have revised the manuscript to address each of the points raised. For detailed information on these changes, please refer to our point-by-point responses to the reviewers' specific comments, which are presented in regular font, following each reviewer comment in bold and italicized font below:

Reviewer 1:

I only had some minor comments. In figure 1, absence is misspelled in the heading. In figure 2A, at least in the review copy pdf, parts of the dendrograms and some of the genome names aren't showing.

RESPONSE: We thank the reviewer for bringing this to our attention. We have corrected the misspelling in figure 1. For figure 2, we suspect that the missing parts were caused by an upload error. We have uploaded a new copy of figure 2, of which we confirmed contains all missing parts.

Another thing I noticed in Fig 2A was there is one genome that is labeled Haemophilus influenzae and sits in the middle of that group in the pangenomics tree but is on a long branch in the phylogenomics tree ending up between parainfluenzae and haemolyticus. I can't see a genome name for that one, but it seems like it might be worth checking to see if there is some kind of error there.

RESPONSE: We agree that the placement of the *H. influenzae* genome (strain NCTC11931; GCA_900475535.1) in Figure 2A warrants further examination.

To address the reviewer's concern, we added the following text to the manuscript:

Lines 240 - 248 "One exception of a single *H. influenzae* genome was notable, where it appears differently positioned in the phylogenomic tree compared to the pangenomic tree. Upon examining the concatenated amino acid sequences for this genome, we discovered that 69 out of the 71 extracted amino acid sequences presented an unusually high number of gaps, leading to its unusual placement between *H. parainfluenzae* and *H. haemolyticus*. This observation underscores the complexities and limitations of relying solely on phylogenomic approaches for determining genomic relationships and highlights the need for incorporating complementary methods, such as pangenomic analyses and average nucleotide identity assessments."

Reviewer 2:

Line 291: Is there a reason you focused on these two isolates in particular? For instance, I would be interested in understanding detailed differences, if any, between the A. aphrophilus in SUPP v SUBP.

RESPONSE: In this part of the manuscript, we concentrated on differences across the major types of oral habitats – dental plaque, tongue dorsum, and buccal mucosa – to underscore the prevalence of a distinct *H. parainfluenzae* subgroup on the tongue dorsum. Interestingly, this subgroup was notably absent or scarcely present in both dental plaque and buccal mucosa samples. On the other hand, *A. aphrophilus* exhibited a significant presence in both SUPP and SUBP samples, without any clear distinction in the distribution patterns between strains of *A. aphrophilus*. This contrast in distribution patterns between *H. parainfluenzae* and *A. aphrophilus* across different oral sites is a key aspect of our findings, highlighting the unique microbial landscapes within the oral cavity.

To address the reviewer’s query, we added the following text to the manuscript:

Lines 299 – 303 “Except for *A. kilianii*, which exhibits a distinct distribution pattern, other taxa that are highly abundant in supragingival plaque, such as *A. aphrophilus*, also tend to show high abundance in subgingival plaque, without any clear distinction in the distribution patterns between strains. This observation suggests adaptation of these taxa to environments that are common to both supra- and sub-gingival plaque.”.

Line 524: Your Github link doesn’t work for me—maybe it’s still private? I can find the user, but nothing listed in the profile that looks like it’s associated with this work.

RESPONSE: We apologize for any inconvenience caused by the GitHub link issue. We have now made the repository publicly accessible and have updated its name to align with our study's title for easier identification. You can find the revised link on lines 550 – 551 of the manuscript. The new URL is:
<https://github.com/FatherofEverest/Spatial-ecology-of-Haemophilus-and-Aggregatibacter-in-the-human-oral-cavity>.

Re: Spectrum04017-23R1 (Spatial ecology of *Haemophilus* and *Aggregatibacter* in the human oral cavity)

Dear Dr. Jonathan J. Giacomini:

Your manuscript has been accepted, and I am forwarding it to the ASM production staff for publication. Your paper will first be checked to make sure all elements meet the technical requirements. ASM staff will contact you if anything needs to be revised before copyediting and production can begin. Otherwise, you will be notified when your proofs are ready to be viewed.

Sincerely,
Justin Kaspar
Editor
Microbiology Spectrum